# An Inverse Kinematics Solution for a Series-Parallel Hybrid Banana-Harvesting Robot Based on Deep Reinforcement Learning

**Guichao Lin** [1,2], **Peichen Huang** [3], **Minglong Wang** [1], **Yao Xu** [1], **Rihong Zhang** [1,*] **and Lixue Zhu** [1,2,*]

1   School of Mechanical and Electrical Engineering, Zhongkai University of Agriculture and Engineering, Guangzhou 510225, China
2   Guangdong Laboratory for Lingnan Modern Agriculture, Guangzhou 510642, China
3   College of Automation, Zhongkai University of Agriculture and Engineering, Guangzhou 510225, China
*   Correspondence: zhangrihong@zhku.edu.cn (R.Z.); zhulixue@zhku.edu.cn (L.Z.)

**Abstract:** A series-parallel hybrid banana-harvesting robot was previously developed to pick bananas, with inverse kinematics intractable to an address. This paper investigates a deep reinforcement learning-based inverse kinematics solution to guide the banana-harvesting robot toward a specified target. Because deep reinforcement learning algorithms always struggle to explore huge robot workspaces, a practical technique called automatic goal generation is first developed. This draws random targets from a dynamic uniform distribution with increasing randomness to facilitate deep reinforcement learning algorithms to explore the entire robot workspace. Then, automatic goal generation is applied to a state-of-the-art deep reinforcement learning algorithm, the twin-delayed deep deterministic policy gradient, to learn an effective inverse kinematics solution. Simulation experiments show that with automatic goal generation, the twin-delayed deep deterministic policy gradient solved the inverse kinematics problem with a success rate of 96.1% and an average running time of 23.8 milliseconds; without automatic goal generation, the success rate was just 81.2%. Field experiments show that the proposed method successfully guided the robot to approach all targets. These demonstrate that automatic goal generation enables deep reinforcement learning to effectively explore the robot workspace and to learn a robust and efficient inverse kinematics policy, which can, therefore, be applied to the developed series-parallel hybrid banana-harvesting robot.

**Keywords:** banana-harvesting robot; series-parallel hybrid robot; inverse kinematics; deep reinforcement learning; twin-delayed deep deterministic policy gradient

## 1. Introduction

According to statistics from the Statistics Bureau of Guangdong [1], the planted area of bananas in Guangdong Province reached 0.11 million hectares, and the output was up to 4.7873 million tons. These bananas are typically picked by the agricultural labor force. Labor shortages and workforce aging are increasing the cost of bananas. Hence, developing a banana-harvesting robot is of great importance to reduce harvesting cost [2]. Each banana cluster has 50 to 150 individual fruits, making the cluster extra heavy. To this end, a series-parallel hybrid robot was developed in our previous work that is capable of clamping the banana stalk. However, the inverse kinematics (IK) problem of this robot is that it is intractable to an address, which is one of the primary challenges facing fruit harvesting robots.

In the agricultural robotics field, the IK problem has been extensively studied. Van Henten et al. [3] designed a seven-degree-of-freedom (DOF) cucumber-harvesting robot and applied a mixed analytic-numerical approach to solve the IK problem. Furthermore, Van Henten et al. [4] reformulated the IK problem as a nonlinear optimization problem and used a genetic algorithm to solve it. Nevertheless, the approach was found to be too

slow for online implementation. Bac et al. [5] developed a nine-DOF sweet pepper-picking robot and calculated the IK using the Gauss–Newton method and Jacobian transpose. Baur et al. [6] derived a closed form solution to the IK for redundant agricultural robots using a local optimization technique, which was utilized by Bac et al. [7] to pick sweet peppers. Silwal et al. [8] applied a dual optimization technique to determine the IK of an apple-harvesting robot. Luo et al. [9] adopted an inverse transformation method to solve the IK of a six-DOF grape-harvesting robot. Lehnert et al. [10] and Arad et al. [11] used an open-source IK library TRAC-IK to address the IK for their sweet pepper-picking robots. Birrell et al. [12] directly employed a built-in IK solution in the robot to move the robot's end-effector to the targets. Most of the methods found in the literature are primarily designed to solve the IK problem of serial robots, and it is unclear whether they are applicable to series-parallel hybrid robots.

In recent years, deep reinforcement learning (DRL) has been widely used in robotics to solve dexterous manipulation tasks, such as reaching random targets and avoiding obstacles. Lillicrap et al. [13] developed a deep deterministic policy gradient (DDPG) to learn policies in continuous action spaces. DDPG showed impressive results in dexterous manipulation. Popov et al. [14] introduced an asynchronous version of DDPG that distributes policy training and data collection across several workers and enables a robot to stack blocks. Our previous work extended DDPG with a recurrent neural network to learn an obstacle avoidance policy for fruit-harvesting robots [15]. Because the value function of DDPG tends to be overestimated, Fujimoto et al. [16] borrowed the idea of double Q-learning and developed a twin-delayed deep deterministic policy gradient (TD3) to limit overestimation. Experiments showed that TD3 greatly outperformed DDPG. DRL learns control policies from exploration, but the amount of exploration required limits its efficiency. To this effect, Vecerik et al. [17] combined DDPG, prioritizing experience replays and expert demonstrations to reduce the exploration problem. Analogously, Nair et al. [18] enhanced DDPG with hindsight experience replay and demonstrations to handle exploration challenges. Schoettler et al. [19] extended TD3 with residual reinforcement learning and demonstrations to improve exploration efficiency. Unfortunately, demonstration data are difficult to obtain by noncooperative robots that do not allow humans to guide the robot to a desired position. Akkaya et al. [20] proposed a novel technique called automatic domain randomization (ADR), which automatically generates randomized environment parameters of increasing difficulty. ADR allowed policies to be trained in simple environments and then improved in difficult ones, thus greatly alleviating the exploration problem.

In short, DRL has great potential for solving the IK problem of the developed series-parallel hybrid banana-harvesting robot. However, the huge workspace of banana harvesting robots may make DRL exploration inefficient. Inspired by ADR, a technique called automatic goal generation (AGG) is developed to generate random targets that are progressively distributed throughout the robot workspace. AGG is combined with a state of the art in DRL, TD3, to learn a robust and efficient IK solution. The main contributions of this work are as follows:

(a) A practical technique called AGG is developed to enable DRL algorithms to explore the robot workspace efficiently.
(b) TD3 is extended with AGG to learn a robust and efficient IK solution for the series-parallel hybrid banana-harvesting robot.
(c) TD3 + AGG achieves impressive results. More specifically, TD3 + AGG greatly outperforms TD3 and obtains a success rate of 96.1% and an average running time of 23.8 milliseconds.

## 2. Materials and Methods

This section proposes a DRL algorithm called TD3 + AGG to learn an IK policy for the series-parallel hybrid banana-harvesting robot that can predict the robot's joint values, given a target. The forward kinematics and workspace of the robot used in the DRL

exploration phase to make the robot perform actions are analyzed in Section 2.1. Section 2.2 illustrates the details of TD3 + AGG. The flowchart of TD3 + AGG is shown in Figure 1, which also shows the relationship between the forward kinematics, robot workspace and TD3 + AGG.

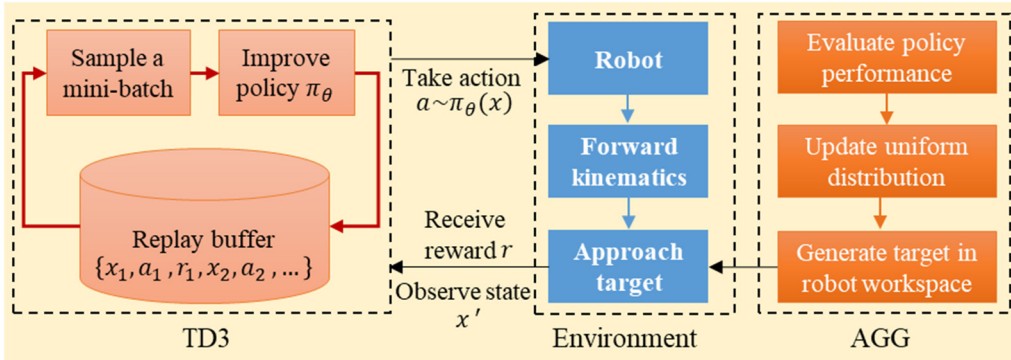

**Figure 1.** Flowchart of TD3 + AGG. TD3 alternately collects transitions through exploration in the robot environment and uses a subset of transitions to optimize the IK policy. Meanwhile, AGG generates random targets for the environment to encourage TD3 to explore the entire robot workspace progressively.

### 2.1. Forward Kinematics and Robot Workspace Analysis

A series-parallel hybrid banana-harvesting robot was previously developed and consists of a waist, a wrist and a manipulator, as shown in Figure 2a. There is one revolute joint at the waist and one revolute joint at the wrist. The manipulator comprises several parallelogram linkage mechanisms, which can not only move the end-effector to the target position but also keep it horizontal. Since banana stalks are usually perpendicular to the ground, a horizontal end-effector would clamp the stalks better. The manipulator has two prismatic joints. Therefore, the robot has four joints and four DOFs. Table 1 lists the joint parameters. Figure 2b shows the kinematic sketch of the robot, where $x_0$-$O_0$-$z_0$ represents the robot base coordinate system (BCS), and $x_{aux}$-$O_{aux}$-$z_{aux}$ represents an auxiliary coordinate system that is fixed on the waist and rotates with the manipulator. The link length is given in Table 2.

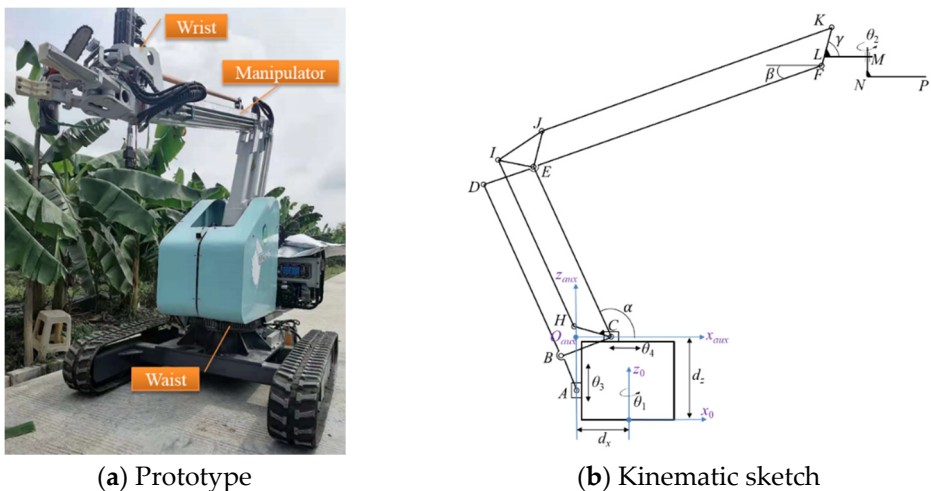

(**a**) Prototype                    (**b**) Kinematic sketch

**Figure 2.** Series-parallel hybrid banana-harvesting robot.

**Table 1.** Joint type and its value range.

| Joint $i$ | Type | Value $\theta_i$ | |
|---|---|---|---|
| | | Initialization | Range |
| 1 | Revolute | 0 | $-\pi/2$ to $\pi/2$ |
| 2 | Revolute | 0 | $-\pi/2$ to $\pi/2$ |
| 3 | Prismatic | $-270$ | $-330$ to $0$ |
| 4 | Prismatic | 360 | 228 to 456 |

**Table 2.** Link length.

| Link | Length (mm) | Link | Length (mm) |
|---|---|---|---|
| $l_{AB}$ | 270 | $l_{EF}, l_{JK}$ | 2080 |
| $l_{BC}, l_{DE}$ | 360 | $l_{FL}$ | 64 |
| $l_{BD}, l_{HI}, l_{CE}$ | 1260 | $l_{LM}$ | 192 |
| $l_{CH}, l_{IE}, l_{JE}, l_{KF}$ | 240 | $l_{MN}$ | 129 |
| $l_{IJ}$ | 354 | $l_{NP}$ | 364 |

The objective of forward kinematics is to compute the end-effector pose, given the robot's joint values. Let $\theta_1$ and $\theta_2$ denote the rotation angles of the two revolute joints, respectively, and let $\theta_3$ and $\theta_4$ denote the movement amounts of the two prismatic joints. The forward kinematics equation is derived according to Zhang et al. [21] as follows:

$$
\begin{cases}
x = \cos\theta_1(\theta_4 + l_{BD}\cos\alpha + l_{EF}\cos\beta + l_{FL}\cos\gamma + l_{LM} - d_x) + \cos(\theta_1 + \theta_2)l_{NP} \\
y = \sin\theta_1(\theta_4 + l_{BD}\cos\alpha + l_{EF}\cos\beta + l_{FL}\cos\gamma + l_{LM} - d_x) + \sin(\theta_1 + \theta_2)l_{NP} \\
z = l_{BD}\sin\alpha - l_{EF}\sin\beta + l_{FL}\sin\gamma - l_{MN} + d_z \\
\varnothing = \theta_1 + \theta_2
\end{cases}
\tag{1}
$$

where $P = (x, y, z)$ and $\varnothing$ are the end-effector position and angle, respectively; $d_x$ (364 mm) and $d_z$ (657 mm) are the horizontal and vertical distances between $x_0$-$O_0$-$z_0$ and $x_{aux}$-$O_{aux}$-$z_{aux}$, respectively; intermediate variables $\alpha$, $\beta$, and $\gamma$ are computed as follows, respectively:

$$
\begin{cases}
\alpha = \arccos\left(\dfrac{l_{AB}^2 + \theta_3^2 + \theta_4^2 - l_{BC}^2}{2l_{AB}\sqrt{\theta_3^2 + \theta_4^2}}\right) + \arccos\left(\dfrac{\theta_4}{\sqrt{\theta_3^2 + \theta_4^2}}\right) \\
\beta = \pi - \arccos\left(\dfrac{l_{AB}^2 + \theta_3^2 + \theta_4^2 - l_{BC}^2}{2l_{AB}\sqrt{\theta_3^2 + \theta_4^2}}\right) - \arccos\left(\dfrac{\theta_4}{\sqrt{\theta_3^2 + \theta_4^2}}\right) - \arccos\left(\dfrac{l_{AB}^2 + l_{BC}^2 - \theta_3^2 - \theta_4^2}{2l_{AB}l_{BC}}\right) \\
\gamma = \dfrac{17}{18}\pi - \arccos\left(\dfrac{l_{IE}^2 + l_{JE}^2 - l_{IJ}^2}{2l_{IE}l_{JE}}\right)
\end{cases}
\tag{2}
$$

It is worth noting that (a) the forward kinematic model is not established based on the Denavit–Hartenberg convention but on trigonometry and algebra [21], and (b) the mobile platform of the robot is omitted in the proposed model.

The robot workspace is approximated by uniformly sampling the joint space of the robot and solving the forward kinematics equation, as shown in Figure 3. Obviously, the boundaries of the workspace can be represented by the following equation:

$$
\begin{cases}
W_P = \{(x, y, z) | -360 \leq x \leq 3100, -3180 \leq y \leq 3180, \\
290 \leq z \leq 2200, 1480 \leq \sqrt{x^2 + y^2} \leq 3180\} \\
W_\varnothing = \{\varnothing | -\pi \leq \varnothing \leq \pi\}
\end{cases}
\tag{3}
$$

This workspace is sufficient for the robot to pick bananas. The usual row spacing for banana plants in China is 2300 mm to 2500 mm. So, the robot workspace in the $y$-axis is larger than it needs to be and is limited to the range of $[-2000, 2000]$ mm, which is enough for the robot to pick bananas.

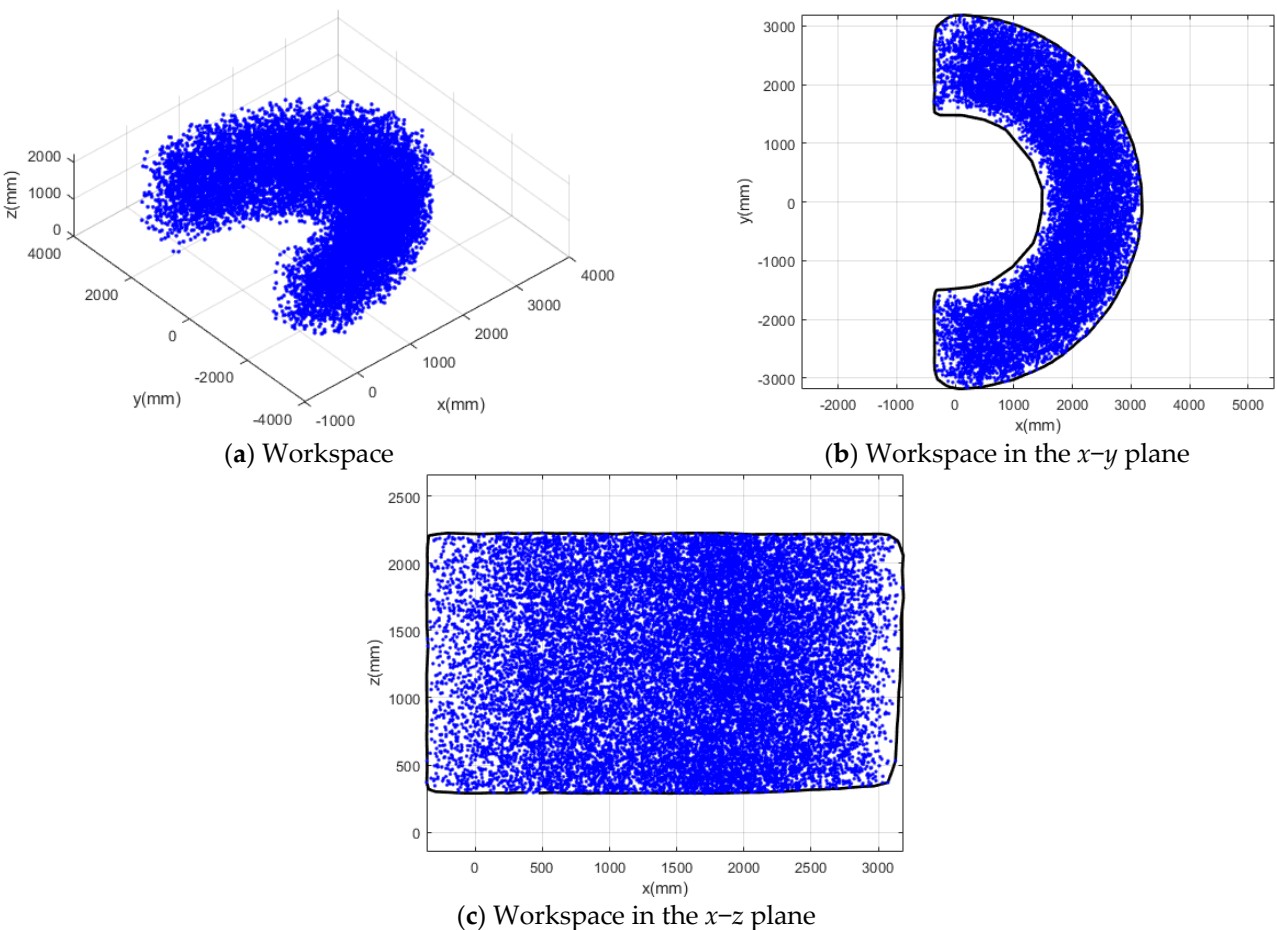

**Figure 3.** Banana−harvesting robot workspace: (**a**) three−dimensional workspace; (**b**) workspace in the *x*–*y* plane; (**c**) workspace in the *x*−*z* plane. Each blue point is obtained by sampling the joint space of the robot randomly and solving the forward kinematics equation. Black lines form the boundaries of the workspace.

*2.2. Inverse Kinematics Analysis*

Equation (1) is a complex nonlinear equation, and thus the analytical solution of the joint values is difficult to derive. Nonlinear programming methods can solve Equation (1) but at the cost of a heavy calculation burden [4]. In pursuit of better speed without loss of accuracy, a DRL-based IK algorithm, called TD3 + AGG, is investigated in this subsection. Section 2.2.1 depicts the background of TD3. Section 2.2.2 introduces a practical AGG technique that enables TD3 to efficiently explore the robot workspace. Section 2.2.3 describes the reward, action and reward components of TD3. The network architecture and learning strategy of TD3 + AGG are detailed in Section 2.2.4.

2.2.1. Preliminaries

The standard Markov decision process is used to model the IK problem. Specifically, at each time-step $t$ in an episode, an agent observes a state $x_t$, takes an action $a_t$, receives a reward $r_t$, and the state evolves into a new state $x_{t+1}$ according to environment transition dynamics. In this study, the episode is equivalent to the IK process of the robot, and the action at the episode end is considered to be a solution to the IK problem. The goal of DRL is to learn a policy $a_t \sim \pi_\theta(x_t)$, with parameters $\theta$, to maximize the expected return $J = \mathrm{E}\left[\sum_{t=1}^{H} \gamma^t r_t\right]$, where $H$ is the episode length and $\gamma$ is a discount factor. The objective $J$ can be maximized by TD3, a state-of-the art-in DRL.

### 2.2.2. Automatic Goal Generation

TD3 requires exploring the entire robot workspace to collect a large number of transitions $(x_t, a_t, r_t, x_{t+1})$ to improve the objective $J$. As the robot workspace is too large, the robot with random exploration rarely obtains a positive reward and always obtains low-quality transitions. The exploration process is therefore inefficient. To this end, a technology called AGG is developed to improve the exploration efficiency. The basic idea of AGG is to train a policy with targets that are progressively distributed throughout the entire robot workspace. AGG is similar to ADR [20], but not exactly the same. AGG only improves target randomness, while ADR increases environmental randomness, such as lighting, friction and gravity. AGG is detailed as follows.

During training, a target is sampled at the beginning of each episode and fixed throughout the whole episode. Specifically, let $\mathcal{G}^0 \in \mathcal{R}^d$ be the initial pose of the end-effector of the robot and $\mathcal{G} \in \mathcal{R}^d$ be a random target. Target $\mathcal{G}$ is sampled form a uniform distribution $\mathcal{G}_i \sim U(\mathcal{G}_i^0 - \psi_i, \mathcal{G}_i^0 + \psi_i)$, $i = 1, \cdots, d$, where $\psi_i$ determines the degree of randomness. Obviously, the larger the value of $\psi_i$, the more random $\mathcal{G}$ becomes. To make TD3 focus on hard tasks, AGG also randomly selects a dimension $k$ and resamples $\mathcal{G}_k$ near its left boundary $[\mathcal{G}_k^0 - \psi_k, \mathcal{G}_k^0 - \psi_k + \Delta_1]$ or right boundary $[\mathcal{G}_k^0 + \psi_k - \Delta_1, \mathcal{G}_k^0 + \psi_k]$, each with a probability of 0.5. Here, $\Delta_1 \geq 0$ is a constant. Then, policy performance is evaluated and appended to a buffer after an episode is finished. Once a training epoch is accomplished, these performances are averaged and compared with a fixed threshold $t$. If the average policy performance is greater than $t$, $\psi_k$ is increased by $\Delta_2$; otherwise, it is decreased by $\Delta_2$, where $\Delta_2 > 0$ is a step size. As seen, TD3 dynamically enlarges the value of $\psi_i$, $i = 1, \cdots, d$, so that the robot can explore its workspace progressively and efficiently.

AGG parameters are updated based on the policy performance. In this work, two kinds of policy performance indicators are investigated, as follows:

(a) Negative $L_2$ distance between the target position and the end-effector position at the episode end, which is defined as $-\|\mathcal{L}_P(\mathcal{G}) - P\|$, where function $\mathcal{L}_P(\mathcal{G})$ returns the position component of target $\mathcal{G}$.

(b) Negatively bounded $L_2$ distance between the target position and the end-effector position at the episode end, which is defined as $-\tan h^2(w_1\|\mathcal{L}_P(\mathcal{G}) - P\|)$, where $w_1$ was set to 0.005 in the experiments.

The pseudocode of AGG is outlined in Algorithm 1. It should be noted that AGG is performed at the beginning of each episode and is only used to generate task targets; $\Delta_1$ and $\Delta_2$ were set to 15 and 30 in experiments, respectively.

---

**Algorithm** 1 AGG

---

**Initialize**: threshold $t$ step size $\Delta_1$ and $\Delta_2$, buffer $D = \varnothing$, robot initial pose $\mathcal{G}^0$, $k = 1$, and sampling randomness $\psi_i$, $i = 1, \cdots, d$.

  **Repeat**:

    **If** an episode is finished:

      Calculate policy performance and append it to $D$

      $\mathcal{G}_i \sim U(\mathcal{G}_i^0 - \psi_i, \mathcal{G}_i^0 + \psi_i)$, $i = 1, \ldots, d$

      $x_1 \sim U(0,1)$, $x_2 \sim U(0,1)$

      **If** $x_1 > 0.5$:

        **If** $x_2 > 0.5$:

          $\mathcal{G}_k \sim U(\mathcal{G}_k^0 - \psi_k, \mathcal{G}_k^0 - \psi_k + \Delta_1)$

        **Else**:

          $\mathcal{G}_k \sim U(\mathcal{G}_k^0 + \psi_k - \Delta_1, \mathcal{G}_k^0 + \psi_k)$

    **If** a training epoch is finished:

      **If** Mean($D$) > $t$:

        $\psi_k = \psi_k + \Delta_2$

      **Else**:

        $\psi_k = \psi_k - \Delta_2$

      $D = \varnothing$, $k \sim U\{1, \ldots, d\}$

---

### 2.2.3. State, Action and Reward

Formally, a state is a set of information observed by the robot. In this study, each state consists of four joint values, the end-effector pose, the target pose, and the relative distance of the end-effector pose to the target pose. These attributes have different value ranges, so they are normalized by subtracting their means and dividing by their standard deviations.

The actions taken by the policy correspond to the robot joint values, since the goal of this study is to solve the IK problem. There are two ways to represent the joint values, as absolute representation or incremental representation. The former has a wide range of variation, so a small network noise would make the robot move too much and therefore destabilizes policy training. For this reason, the incremental representation is used. In our experiments, the range of the incremental values of the revolute joints was set to $\left[-2^{\circ}, 2^{\circ}\right]$ and that of the prismatic joints was set to $[-2\,\text{mm}, 2\,\text{mm}]$.

The rewards are used to measure the returns of state–action pairs. Sparse reward requires little domain-specific knowledge and is easy to specify, while dense reward requires domain-specific knowledge to encourage the robot to accomplish its target and is slightly intractable to define. These two kinds of reward are investigated here.

(a) Sparse reward: the robot receives a positive reward if the target is reached and a negative reward otherwise.

$$r(s,a) = \begin{cases} 1 & if \; \|\mathcal{L}_P(\mathcal{G}) - P\| \leq \varepsilon \\ -1 & otherwise \end{cases} \tag{4}$$

where $\varepsilon$ was set to 20 mm.

(b) Dense reward: this reward comprises a position-based reward [14], an angle-based reward, and a bonus when the target is reached.

$$r(s,a) = -\tanh^2(w_1\|\mathcal{L}_P(\mathcal{G}) - P\|) + w_2(\cos(\mathcal{L}_\varnothing(\mathcal{G}) - \varnothing) - 1) + I_{\{\|\mathcal{L}_P(\mathcal{G})-P\|\leq\varepsilon\}} \tag{5}$$

where function $\mathcal{L}_\varnothing(\mathcal{G})$ returns the angle component of target $\mathcal{G}$; $I_{\{\|\mathcal{L}_P(\mathcal{G})-P\|\leq\varepsilon\}}$ is an indicator function that returns 1 if $\|\mathcal{L}_P(\mathcal{G}) - P\| \leq \varepsilon$, and 0 otherwise; weights $w_1$ and $w_2$ were set to 0.005 and 0.02, respectively.

### 2.2.4. Learning Inverse Kinematics Policy

Network Architecture

The policy is trained with TD3, which requires the training of the following two networks: a policy network, which maps a state to an action, and a value network, which predicts a discounted sum of future rewards for a state–action pair. Both networks consist of a normalization layer and two fully connected hidden layers with 100 units each. Each hidden layer is activated by the ReLU6 function. The layer weights are initialized using a random orthogonal initialization method [22], and there is no layer sharing between the two networks. For the policy network, the output is activated by the softsign activation function and scaled to the range of the incremental joint value. The network architectures are shown in Figure 4.

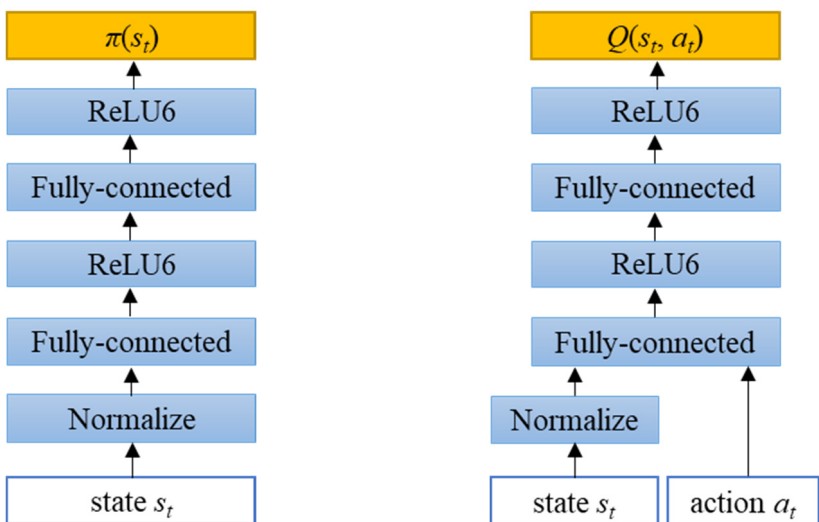

**Figure 4.** Policy network (**left**) and value network (**right**).

Training Strategy

The two network parameters are learned by the Adam optimizer [23] with a learning rate of $10^{-3}$. A total of 1500 epochs with 200 steps each are used. The discount factor, mini-batch size, episode length and replay buffer size are set to 0.95, 200, 200 and $10^5$, respectively. To stabilize network learning, $L_2$ regularization on the two network parameters is added to the objective of TD3. For the exploration process, Gaussian noise with a mean of 0 and a standard deviation of 0.5 are added to the actions. All the transitions are gathered in a simulation environment, which only contains the series-parallel hybrid banana harvesting robot and the targets. Figure 5 shows the environment developed by using the Python library pyglet.

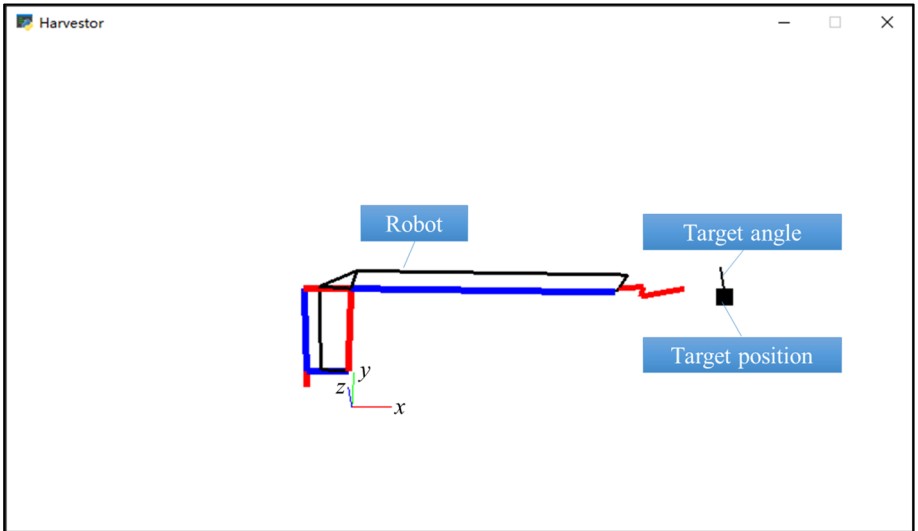

**Figure 5.** Simulation environment.

### 2.3. Experimental Setups

Simulation and field experiments are performed to evaluate the performance of the learned IK policy of TD3 + AGG. TD3 + AGG and the image processing algorithm are programmed using TensorFlow2 and Opencv3, and the comparison algorithm is programmed by MATLAB R2021a. All the codes are run on a computer with a Windows 10 system, 32 GB of RAM, and an Intel i7-10700K CPU.

2.3.1. Simulation Experiments

The objective of the simulation experiments is to answer the following questions: can AGG improve the exploration efficiency and enable TD3 to solve the IK problem for the series-parallel hybrid robot? Can TD3 + AGG outperform a traditional nonlinear programming approach? Three experiments are performed and detailed below.

(a) Exploration with AGG. It is unclear whether the AGG can enable TD3 to explore the entire robot workspace effectively. In addition, the AGG parameters are updated based on the policy performance, and it is not clear which performance indicator is the best. To this end, this experiment uses the original TD3 as a baseline and evaluates the success rates of TD3 + AGG with different performance indicators on one thousand random targets generated in the robot workspace. It should be noted that an IK solution is considered successful only if the distance between the end-effector and the target is below a threshold, which was set to 20 mm in the experiments; and the success rate is calculated as the ratio of the number of successful solutions to the total number of targets.

(b) Learning from different rewards. Two reward functions are studied: sparse reward and dense reward. The experiment uses TD3 + AGG with the two rewards as the learning algorithms, performs one thousand picking attempts in the robot workspace, and then analyses their success rates.

(c) Comparison with a nonlinear programming approach. The IK problem is reformulated as a nonlinear optimization problem, as formulated in Equation (6) [4], which is minimized by using the *fmincon* function from MATLAB R2021a to find a solution. The experiment generates one thousand random targets in the robot workspace, implements the nonlinear programming approach and TD3 + AGG to solve the IK for each target, and then analyzes the pose errors, running times and success rates.

$$C(\theta) = \|\mathcal{L}_P(\mathcal{G}) - P(\theta)\| + \eta\|\mathcal{L}_\varnothing(\mathcal{G}) - \varnothing(\theta)\| \tag{6}$$

where $\theta$ represents the joint values of the robot and $\eta$ is a balance factor, which was set to 20 in the experiments.

2.3.2. Field Experiment

Experimental Setup

The objective of the field experiment is to evaluate IK policy performance. The field experiment was conducted in a commercial banana orchard in Guangzhou, China, on 21 and 22 September 2021. An end-effector is customized to grip and cut the banana stalk, which has relatively large tolerances in its width and depth directions, as shown in Figure 6. The end-effector works as follows: it first grips the stalk using its fingers at a moving speed of 5 mm/s, and then cuts the stalk using a swing chainsaw whose swing speed is somewhat slow for avoiding vibration. Because our previous experiments found that each banana cluster weighed 40~60 kg and required a force of 848~1322 N for a stable grip, the maximum griping force was used to grip the stalk in the experiments to avoid sliding. Great details about the end-effector can be found in our patent (CN211406875U). In the experiment, the robot automatically moves along the path and scans the stalk. Once a stalk is detected, the robot stops moving, uses the IK policy to calculate the joint values, and then approaches the stalk. A total of 50 picking attempts were performed, and the positioning error of the end-effector center with respect to the stalk center was measured. Figure 7 displays how the error is measured.

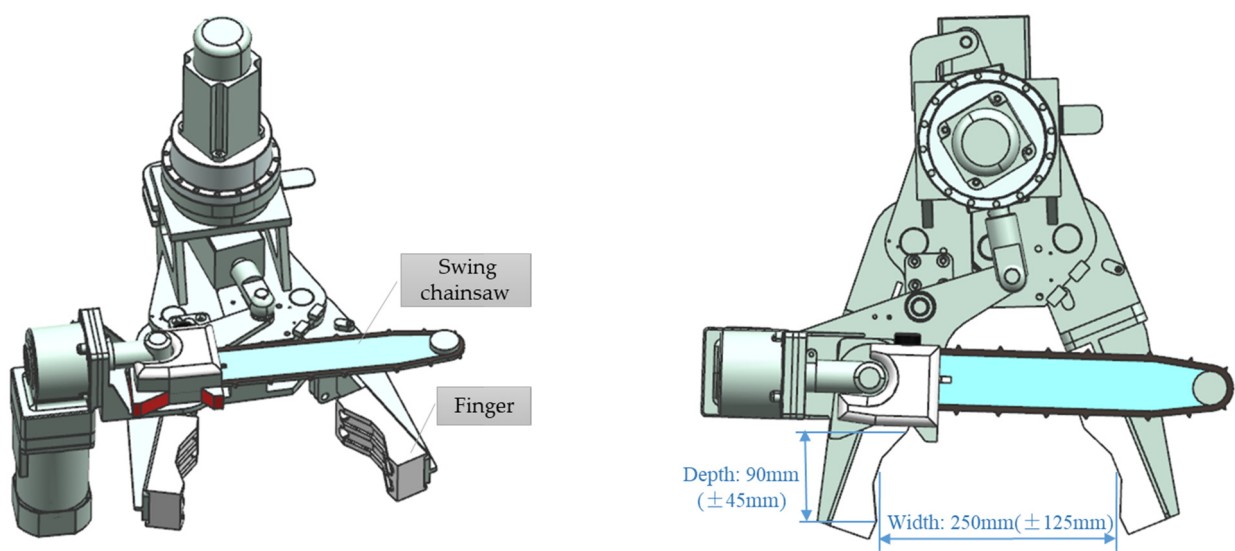

**Figure 6.** End-effector (**left**) and its tolerances in the width and depth directions (**right**).

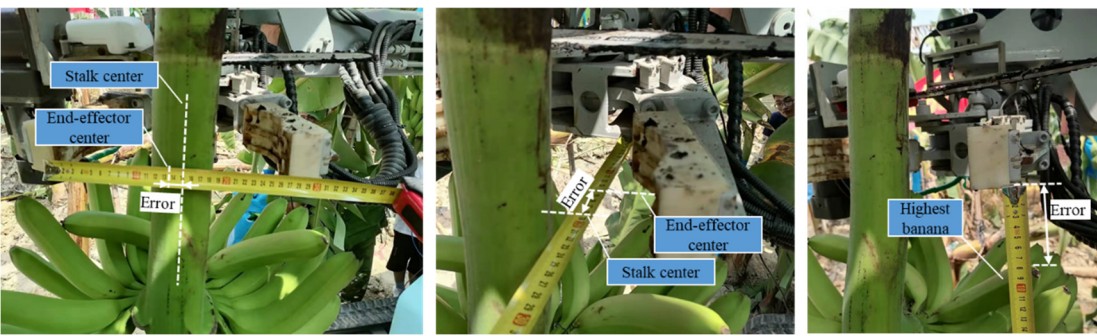

**Figure 7.** Example showing how to measure the errors of the end-effector relative to the stalk center in the end-effector width (**left**), depth (**middle**) and height (**right**) directions. Notably, because the actual height of a detected stalk is unknown, the height error is determined by measuring the vertical distance between the end-effector and the top fruit.

Image Processing Pipeline

During the field experiment, the pose of the banana stalk needs to be determined. The methods developed in our previous studies were deployed to detect and locate banana stalks [24,25]. Specifically, a depth filter is first used to remove distant objects in an image, and then a fully convolutional neural network is implemented to segment the filtered image to obtain a banana stalk region (Figure 8a). Afterward, the banana stalk region is converted into a point cloud, and a principal component analysis-based cylinder-fitting algorithm is performed on the point cloud to fit a cylinder (Figure 8b). The cylinder center is regarded as the stalk position. From the robot point of view, the stalk is usually in front of the pseudostem. Therefore, the stalk angle is simply set to $\pi/2$ if the $y$ value of the stalk is positive and $-\pi/2$ otherwise to guarantee that the end-effector can approach the stalk without collision with the pseudostem. To enable the robot to "see" the stalk, the stalk position is mapped from the camera coordinate system (CCS) to BCS. The relationship between CCS and BCS is determined by a hand-eye calibration method [26]. It should be noted that low-cost depth camera RealSense D4335i was used for image acquisition, which can generate a pair of aligned RGB and depth images at 60 frames per second with a resolution of $640 \times 480$ pixels.

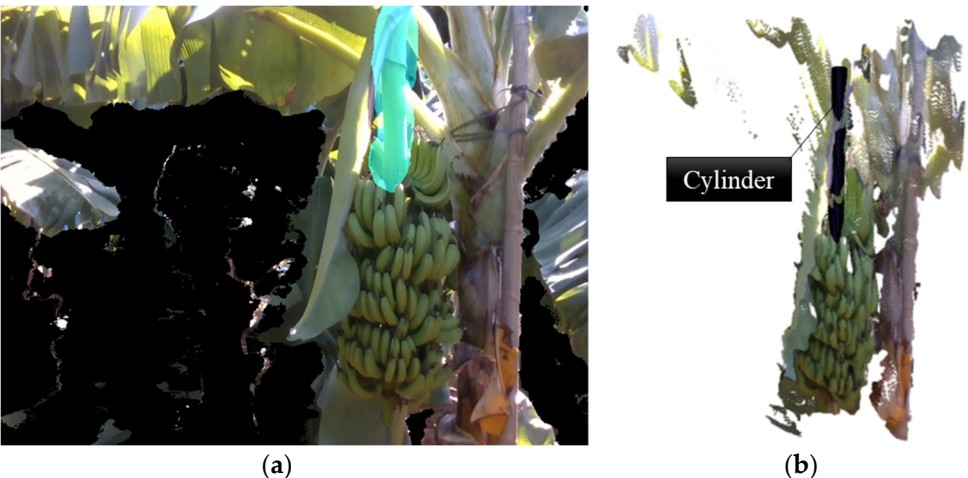

**Figure 8.** Example depicting the image processing pipeline. (**a**) Banana stalk segmentation; (**b**) cylinder fitting.

## 3. Results and Discussion

### 3.1. Simulation Experiments

#### 3.1.1. Exploration with AGG

Table 3 lists the success rates of TD3 and different variants of TD3 + AGG. With sparse reward, TD3 was unable to learn to reach the targets, while TD3 + AGG surprisingly learned to approach most targets. With a dense reward, TD3 + AGG had a more encouraging result than TD3. These results demonstrate that AGG allowed TD3 to explore the robot workspace progressively and thereby made learning effective even with a sparse reward.

**Table 3.** Success rates of TD3 and different variants of TD3 + AGG.

| Algorithm | AGG | | Reward Function | | Success Rate |
|:---:|:---:|:---:|:---:|:---:|:---:|
| | Negative $L_2$ Distance | Negative-Bounded $L_2$ Distance | Sparse Reward | Dense Reward | |
| TD3 | | | ✓ | | 0.1% |
| TD3 | | | | ✓ | 81.2% |
| TD3 | ✓ | | ✓ | | 80.0% |
| TD3 | ✓ | | | ✓ | 92.9% |
| TD3 | | ✓ | ✓ | | 83.9% |
| TD3 | | ✓ | | ✓ | 96.1% |

AGG performance is affected by the policy performance indicator. Table 3 shows that the negative-bounded $L_2$ distance was superior to the negative $L_2$ distance in terms of success rate.

#### 3.1.2. Learning from Different Rewards

As described in Table 3, the variants of TD3 + AGG with dense reward greatly outperformed those with sparse reward. This indicates that a dense reward was more suitable for guiding the robot to learn to solve the IK problem. The dense reward investigated in this work may not be optimal, and hence, a better dense reward can be shaped to further boost the performance of TD3 + AGG in future work.

Notably, because the negative-bounded $L_2$ distance and dense reward improved the performance of TD3 + AGG the most, the following simulation and field experiments use TD3 + AGG with negative-bounded $L_2$ distance and dense reward as the IK solver.

### 3.1.3. Comparison with a Nonlinear Programming Approach

Table 4 shows the success rates and running times of TD3 + AGG and a nonlinear programming approach. TD3 + AGG slightly outperformed the nonlinear programming approach in terms of success rate. Additionally, TD3 + AGG was computationally more efficient, likely because TD3 + AGG used a tiny neural network with only two fully connected layers for inference, while the nonlinear programming approach required many iterations to converge to a reasonable solution. The result makes clear that TD3 + AGG was able to address the IK problem for the series-parallel hybrid robots.

**Table 4.** Success rates and running times of TD3 + AGG and a nonlinear programming approach.

| Algorithm | Success Rate | Running Time (Milliseconds) | |
| --- | --- | --- | --- |
| | | Mean | Standard Deviation |
| TD3 + AGG | 96.1% | 23.8 | 19.8 |
| Comparison algorithm | 95.0% | 38.0 | 13.3 |

The mean and standard deviation of the pose errors between the end-effector and targets for the two algorithms were also analyzed, as depicted in Table 5. The mean and standard deviation errors in the $x$, $y$, $z$, and angle axis of TD3 + AGG were slightly larger than those of the nonlinear programming approach. Even so, these errors were far less than the end-effector tolerances (see Figure 6). Therefore, the accuracy of TD3 + AGG was sufficient for our series-parallel hybrid robot.

**Table 5.** Mean and standard deviation of pose errors between end-effector and targets.

| Algorithm | Mean | | | | Standard Deviation | | | |
| --- | --- | --- | --- | --- | --- | --- | --- | --- |
| | $x$ (mm) | $y$ (mm) | $z$ (mm) | Angle (Radian) | $x$ (mm) | $y$ (mm) | $z$ (mm) | Angle (Radian) |
| TD3 + AGG | 0.32 | −1.93 | 6.26 | −0.01 | 11.48 | 11.52 | 8.01 | 0.73 |
| Comparison algorithm | 0.97 | −0.21 | 0.11 | −0.01 | 7.39 | 7.24 | 2.01 | 0.36 |

The target positions not reached by the robot were plotted, as shown in Figure 9. Both algorithms struggled to reach the boundary of the robot workspace. Therefore, if a banana stalk is close to the boundary, the robot can be moved forward or backward so that the banana stalk is in the middle of the robot workspace. Furthermore, TD3 + AGG failed to approach a few targets that were not near the boundary. This problem reveals a shortcoming of TD3 + AGG, that the learned IK policy might have little uncertainty.

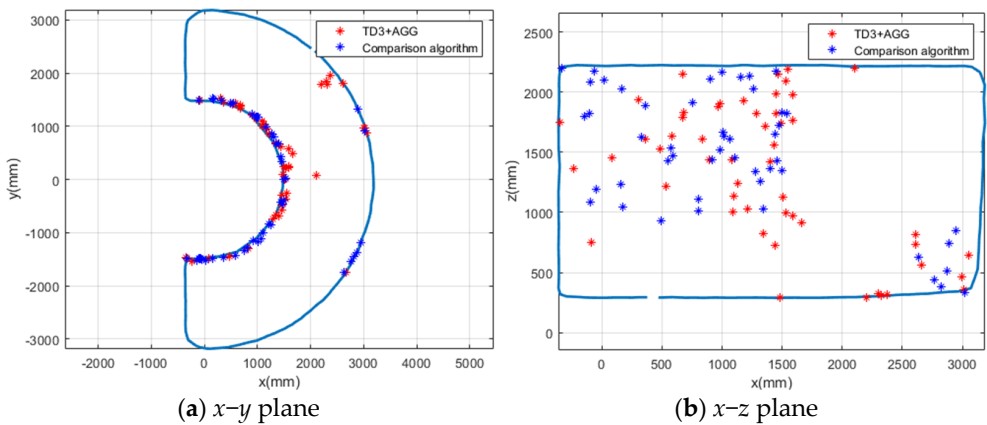

**(a)** $x-y$ plane          **(b)** $x-z$ plane

**Figure 9.** Positions of the targets that are not reached by the robot.

### 3.1.4. Discussion

Sparse reward is attractive because it is easy to specify, but exploration with sparse reward is extremely difficult. Experiments showed that AGG could promote TD3 to explore the robot workspace with sparse reward and yielded an encouraging result. Nonetheless, there is still much room for AGG to improve DRL with sparse reward. Hindsight experience replay and demonstration have been successfully used for accelerating exploration in environments with sparse reward [18,27], which could be combined with AGG in future work.

Overall, the pose accuracy of TD3 + AGG met the requirement of the series-parallel hybrid robot, but it was slightly low. Our previous research also reported a similar result [15]. A possible reason was that the objective of DRL was to maximize a long-term return, not the distance between the robot and the target, which made it difficult for DRL to solve the IK problems accurately. This problem could be alleviated by narrowing the output range of the policy network at the cost of increased inference time.

### 3.2. Field Experiment

The experimental results showed that the IK policy enabled the robot to successfully reach all banana stalks. Furthermore, the positioning errors in the end-effector width, depth and height directions were measured, as listed in Table 6. The errors in the width and depth directions were 5 mm $\pm$ 17 mm and 5 $\pm$ 12 mm, respectively. Because the end-effector has tolerances of $\pm$125 mm and $\pm$45 mm in the width and depth directions, respectively, as shown in Figure 6, the width and depth errors were within the tolerance range and thus acceptable. The error in the height direction was 60 mm $\pm$ 32 mm. Fortunately, the height error was positive and thus prevented the robot from colliding with the fruits. Figure 10 shows the field experiment scenario.

**Table 6.** Mean and standard deviation of the positioning error of the end-effector center with respect to the stalk center.

| Direction | Mean (mm) | Standard Deviation (mm) |
|-----------|-----------|-------------------------|
| Width | 5 | 17 |
| Depth | 5 | 12 |
| Height | 60 | 32 |

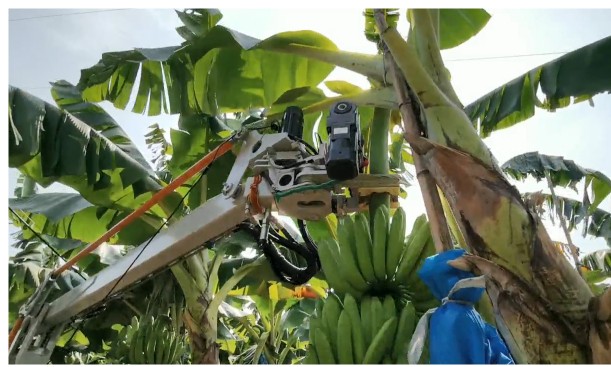 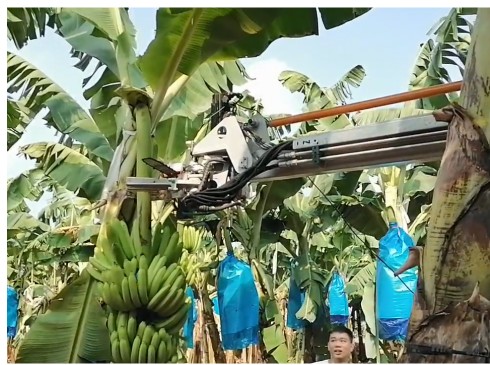

**Figure 10.** Field experiment scenario.

Discussion

The error in the end-effector height direction was determined by measuring the vertical distance between the end-effector and the top fruit for convenience. That is, the measurement did not reflect the actual error between the end-effector and the target. Future research will try to measure the actual height error to evaluate the IK policy performance more objectively.

The angle of the banana stalk relative to its pseudostem was set to $\pi/2$ or $-\pi/2$ This setting simplified the developed image-processing algorithm. However, it was found through experiments that there were a few cases where the robot almost collided with the pseudostem because the actual stalk angle was not always $\pi/2$ or $-\pi/2$. Estimating a reasonable fruit picking angle would reduce the likelihood of collisions, which has attracted the attention of some researchers [28,29]. Our future work will attempt to address the picking angle estimation problem.

### 4. Conclusions

This study investigated a DRL to handle the IK problem of the developed series-parallel hybrid banana harvesting robot. It was found that the method can accomplish the research objective. Some specific conclusions drawn from the study were given as follows:

(a)　In order to encourage DRL to explore the robot workspace efficiently, a novel AGG technique was proposed, which automatically generates random targets with increasing randomness. AGG was combined with TD3 to solve the IK problem for the banana-harvesting robot. Simulation experiments showed that TD3 + AGG solved the IK problem with a success rate of 96.1% and an average running time of 23.8 milliseconds and outperformed a nonlinear programming approach in terms of success rate and running time. It was found that the pose error of TD3 + AGG was somewhat large although within the end-effector tolerance.

(b)　To implement TD3 + AGG on the developed banana-harvesting robot, image processing was introduced. It first uses a fully convolutional neural network to segment the banana stalk and then applies a cylinder-fitting method to fit the stalk point cloud. The center of the resulting cylinder is regarded as the banana stalk position. The angle of the banana stalk relative to its pseudostem was set to $\pi/2$ or $-\pi/2$ for convenience, which however did not reflect the actual stalk angle and might cause the robot to collide with the pseudostem.

(c)　A total of 50 picking attempts were performed in the fields. The experimental results showed that the learned policy of TD3 + AGG solved the IK tasks successfully and enabled the banana-harvesting robot to reach all the banana stalks quite accurately.

In summary, TD3 + AGG is able to solve the IK problem robustly and efficiently and therefore can be applied to the series-parallel hybrid banana-harvesting robot. Future work will mainly focus on improving the positioning accuracy of TD3 + AGG and estimating the picking angle of the stalk relative to the pseudostem.

**Author Contributions:** Conceptualization, G.L. and P.H.; methodology, G.L.; software, M.W. and Y.X.; validation, G.L., M.W. and Y.X.; formal analysis, P.H.; investigation, G.L.; resources, R.Z.; data curation, M.W. and Y.X.; writing—original draft preparation, G.L.; writing—review and editing, R.Z. and L.Z.; visualization, G.L.; supervision, L.Z.; project administration, R.Z.; funding acquisition, G.L., P.H. and L.Z. All authors have read and agreed to the published version of the manuscript.

**Funding:** This work was funded by the Laboratory of Lingnan Modern Agriculture Project (Grant No. NZ2021038), the National Natural Science Foundation of China (Grant No. 32101632), the Basic and Applied Basic Research Project of Guangzhou Basic Research Plan (Grant No. 202201011310; 202201011691), and the Science and Technology Program of Meizhou, China (Grant No. 2021A0304004).

**Data Availability Statement:** Data recorded in the current study are available in all tables and figures of the manuscript.

**Conflicts of Interest:** The authors declare no conflict of interest.

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
