# Peer review of "An Inverse Kinematics Solution for a Series-Parallel Hybrid Banana-Harvesting Robot Based on Deep Reinforcement Learning"

_agronomy, doi:10.3390/agronomy12092157_

Round 1

Reviewer 1 Report

The comments and suggestions are in the attached file of the manuscript

Reviewer 2 Report

Agricultural picking and harvesting robots are a very hot and promising field. This manuscript investigated the inverse kinematics solution method of the banana-harvesting robot by the deep reinforcement learning. The research topic is interesting, but the research information is rough and it is not a high quality study. Hopefully, the authors can improve their research according to the following suggestions.

1. As shown by the Figure 2 and Figure 5, Although the authors claimed the mechanisam is a series-parallel hybrid type, the presented mechanisam is not complex. In robotics, contrary to serial linked type, it is easier for parallel robots to solve its inverse kinematics solution. Therefore, the inverse kinematics solution of the series-parallel hybrid mechanisam as given also have analytical solutions. While the authors seeked to the deep reinforcement learning method, this is not a good option. For the whole proces of Identification, positioning and grabbing, the inverse kinematics is only a small and basic part for the banana-harvesting robot. It is unwise to apply the method of deep learning only in this part (inverse kinematics).

2. As for the accuracy of the inverse kinematics solution as given, the reseach doesn't need to be in a real place, because there are too many uncontrollable influencing  factors, but should be on a more scientific measurement platform. That is to say, the research questions and demonstration experiments are not scientific configurations.I hope to see more detailed and accurate research design of problems, methods and applications.

3. The size and specific parameter values of the mechanism should also be given clearly.

Reviewer 3 Report

Excellent work, in trying to solve one of the greatest problems modern agriculture is going to face on the following years. Just some minor comments to further improve the quality of the manuscript.

Line 31: Please covert mu unit of measurements to hectares as that is the most common measurement unit used worldwide

Line 36: You are referring to figure 2 and not one, please correct it.

Line 75: Please correct reference number 20. It should be Akkaya et al and not Openai. Do the same for the reference list using the following:

Akkaya, Ilge, Marcin Andrychowicz, Maciek Chociej, Mateusz Litwin, Bob McGrew, Arthur Petron, Alex Paino et al. "Solving rubik's cube with a robot hand." arXiv preprint arXiv:1910.07113 (2019).

Line 363: Please define more precisely what you mean by success rate.

Finally, I would also suggest adding in the text the type of sensors used as they are important when we are talking about machine vision.

Round 2

Reviewer 2 Report

1. It's better to use D-H table to show the parameters of the robots.

2. As for the nonlinear solutions, the numerical method maybe is better than the deep learning method. As you know, the nonlinear is common in the robotic kinematics.
